# Contribution of p38 MAPK Pathway to Norcantharidin-Induced Programmed Cell Death in Human Oral Squamous Cell Carcinoma

**DOI:** 10.3390/ijms20143487

**Published:** 2019-07-16

**Authors:** Chi-Hyun Ahn, Kyoung-Ok Hong, Bohwan Jin, WonWoo Lee, Yun Chan Jung, Hakmo Lee, Ji-Ae Shin, Sung-Dae Cho, Seong Doo Hong

**Affiliations:** 1Department of Oral Pathology, School of Dentistry and Dental Research Institute, Seoul National University, Seoul 03080, Korea; 2Laboratory Animal Center, CHA University, CHA Biocomplex, Sampyeong-dong, Seongnam 13488, Korea; 3Chaon, 301-3, 240, Pangyoyeok-ro, Bundang-gu, Seongnam 13493, Korea; 4Veterans Medical Research Institute, Veterans Health Service Medical Center, Seoul 05368, Korea

**Keywords:** norcantharidin, oral squamous cell carcinoma, p38 MAPK, programmed cell death

## Abstract

Norcantharidin (NCTD), a demethylated analog of cantharidin isolated from blister beetles, has been used as a promising anticancer agent; however, the underlying function of NCTD against human oral squamous cell carcinoma (OSCC) has not been fully understood. Here, this study was aimed to investigate the apoptotic effect and molecular targets of NCTD in human OSCC in vitro and in vivo. The anticancer effects of NCTD and its related molecular mechanisms were evaluated by trypan blue exclusion assay, live/dead assay, western blotting, 4-6-Diamidino-2-Phenylindole (DAPI) staining, flow cytometric analysis, Terminal Deoxynucleotidyl Transferase dUTP Nick end Labeling (TUNEL) assay, and immunohistochemistry. NCTD significantly inhibited cell growth and increased the number of dead cells in HSC-3 and HN22 cell lines. It induced the following apoptotic phenomena: (1) the cleavages of poly (ADP-ribose) polymerase and casepase-3; (2) increase in apoptotic morphological changes (nuclear condensation and fragmentation); (3) increase in annexin V-positive cells or sub-G_1_ population of cells. NCTD significantly activated the p38 mitogen-activated protein kinase (MAPK) pathway but inactivated the signal transducer and activator of transcription (STAT)3 pathway. A p38 MAPK inhibitor (SB203580) partially attenuated NCTD-induced programmed cell death (apoptosis) in both cell lines, whereas ectopic overexpression of STAT3 did not affect it. NCTD strongly suppressed tumor growth in the tumor xenograft bearing HSC-3 cells, and the number of TUNEL-positive cells increased in NCTD-treated tumor tissues. In addition, NCTD did not cause any histopathological changes in the liver nor the kidney. NCTD induced programmed cell death via the activation of p38 MAPK in OSCC. Therefore, these results suggest that NCTD could be a potential anticancer drug candidate for the treatment of OSCC.

## 1. Introduction

Oral Squamous cell carcinoma (OSCC) is a major type of head and neck squamous cell carcinoma (HNSCC) (more than 90%) and remains the main cause of HNSCC-associated morbidity and mortality [1]. The most widely used treatments for managing OSCC include surgery, radiation therapy, and chemotherapy, or any combination of all three [2]. Despite their conceptual promise, more than a third of OSCC patients are often untreatable [3]. In particular, the unchanging survival in patients with OSCC underscores the need for better prognostic tools, such as the adding of “Depth” and “Extranodal extension” in the 8th Edition of American Joint Committee on Cancer staging system, as recently reported [4]. Thus, finding signaling pathways underlying molecular basis of OSCC carcinogenesis and developing new drugs targeting them are required.

Several alterations of oncogenic intracellular signaling pathways, including mitogen-activated protein kinase (MAPK), phosphatidylinositol-3-kinase (PI3K)/AKT/mammalian target of rapamycin (mTOR), and signal transducer and activator of transcription (STAT) pathways, contribute to the malignant growth and metastatic potential of OSCC [5]. Among them, STAT3 is strongly associated with OSCC development and progression [6]. Indeed, several evidences demonstrated that aberrant activation of STAT3 has been correlated with cell cycle deregulation, enhanced proliferation, and evasion of programmed cell death, as well as adverse clinical parameters in OSCC [7,8]. Also, dysregulation of p38 MAPK, one of the main sub-group of MAPKs, is a frequent event and has been implicated in advanced stages and short survival of cancer patients [9]. Some studies have suggested that p38 MAPK is highly accumulated in most OSCC lesions [10,11]. By contrast, some authors have concluded that the activation of p38 MAPK inhibits cell proliferation and cancer cell migration in oral cancer [12,13,14]. However, the role of p38 MAPK and STAT3 in OSCC is still obscure.

Natural products derived from a variety of sources, such as plants, animals, and microorganisms, have been considered as new chemotherapeutic agents with fewer side effects [15,16]. In oral cancer research, several natural products show the potential of alternative treatments, as well as complementary treatments, to use in conjugation with chemotherapy, radiation, and surgery [17,18]. Norcantharidin (NCTD), a synthetic demethylated analog of cantharidin isolated from mylabris (blister beetles), has been reported to possess a potent anticancer activity and relatively low intrinsic toxicity in comparison with cantharidin [19]. Previous studies have indicated that NCTD exerts anticancer activity against numerous cancer cell lines. For example, NCTD inhibited invasion ability and metastasis in colorectal cancer cells and breast cancer cells [20,21], induced cell death via mitochondrial-dependent programmed cell death and autophagy in prostate cancer cells [22,23], and exhibited anti-angiogenic activity in gallbladder cancer cells [24]. However, the anticancer effect of NCTD in OSCC still remains unclear.

In the present study, we investigated in vitro and in vivo anticancer activity and possible signaling pathways of NCTD against OSCC.

## 2. Results

### 2.1. NCTD Inhibits Cell Viability and Induces Cell Death in Human OSCC Cell Lines

To identify the anti-proliferative function of NCTD, HSC-3, HN22, and HSC-4 cells were treated with various concentrations of NCTD (0, 15, or 30 μM) for 48 h or 30 μM NCTD for 12, 24, or 48 h. To assess the effect of NCTD on cell viability, the trypan blue exclusion assay was performed. As shown in Figure 1A,B, NCTD significantly reduced cell growth in a concentration- and time-dependent manner. HSC-3 and HN22 cells showed maximum growth inhibition of approximately 46–47% at the highest concentration of NCTD. In line with these results, HSC-4 cells also showed the decrease in cell viability compared with the vehicle control group (Appendix A). To explore whether the growth inhibition by NCTD was associated with cell death, a live/dead staining was performed. This assay is based on the simultaneous determination of live and dead cells with two probes that measure recognized parameters of cell viability-intracellular esterase activity and plasma membrane integrity [25]. As illustrated in Figure 1C and Appendix A, NCTD treatment led to an increase in the ratio of dead cells stained by EthD-1 (red color) in a concentration-dependent manner. Collectively, these results demonstrated that NCTD can promote inhibition of cell growth and induce programmed cell death in OSCC cell lines.

### 2.2. NCTD Induces Programmed Cell Death (Apoptosis) in OSCC Cell Lines

To test whether programmed cell death was affected by NCTD, cleaved poly (ADP-ribose) polymerase (PARP) and cleaved caspase 3 as apoptosis markers were determined using western blotting. The cleavages of caspase 3 and PARP were markedly induced in NCTD-treated OSCC cell lines (Figure 2A and Appendix A). We confirmed NCTD-mediated programmed cell death using DAPI staining, annexin V/PI double staining, and cell cycle analysis. As depicted in Figure 2B,E, and Appendix A, NCTD-treated OSCC cell lines showed apoptotic nuclei with condensation or fragmentation (white arrow indicated). Annexin V/PI double staining showed that the number of both HSC-3 and HN22 cells undergoing early-stage apoptosis (annexin V+/PI−) and late-stage apoptosis (annexin V+/PI+) was significantly increased, whereas almost no apoptotic cells were detected in vehicle control group (Figure 2C,F). In addition, the population of cells in sub-G_1_ phase (apoptosis region) significantly increased in a concentration-dependent manner (Figure 2D,G). Taken together, these results showed that NCTD promotes in vitro programmed cell death in human OSCC.

### 2.3. p38 MAPK is Involved in NCTD-Induced Programmed Cell Death in OSCC Cell Lines

Oncogenic intracellular signaling pathways have been well characterized and are considered as significant OSCC promoting factors [5]. To understand the underlying mechanism of NCTD-induced programmed cell death, we evaluated the effects of NCTD on oncogenic intracellular signaling pathways, including p38 MAPK, STAT3, AKT, extracellular signal-regulated kinase (ERK), and mTOR. As shown in Figure 3, NCTD significantly induced the activation of p38 MAPK at all of time points, and NCTD markedly decreased the phosphorylation of STAT3 compared to the vehicle control group. However, NCTD showed no apparent effect on the activation of AKT, ERK, and mTOR. These results indicate that p38 MAPK and STAT3 may be involved in NCTD-induced programmed cell death in human OSCC cell lines. Thus, we postulated that the inactivation of p38 MAPK or over-expression of STAT3 may recover from NCTD-induced programmed cell death. To ascertain the involvement of p38 MAPK or STAT3 in NCTD-induced anticancer activity in human OSCC cell lines, both cell lines were pretreated with a p38 MAPK inhibitor (SB203580) for 1 h or transiently transfected with STAT3 over-expression vector for 24 h, followed by NCTD treatment for 48 h. SB203580 significantly reversed the suppression of cell growth and PARP cleavages mediated by NCTD (Figure 4A,B). In agreement with these findings, Figure 4C,D showed that treatment of SB203580 significantly reduced the effect of NCTD-mediated programmed cell death, evidenced by the increases in the number of annexin V-positive cells and sub-G_1_ population. On the other hand, the forced expression of STAT3 did not attenuated NCTD-mediated PARP cleavages in both cell lines (Appendix A). These data suggest that the activation of p38 MAPK is a key signaling pathway in NCTD-induced programmed cell death in human OSCC cell lines.

### 2.4. NCTD Inhibits the Xenograft Tumor Growth of Human OSCC by Inducing Programmed Cell Death

To investigate the in vivo anti-tumor activity of NCTD, we subcutaneously injected HSC-3 cells into the flank of athymic nude mice, and low or high doses of NCTD (2.5 or 5 mg/kg) were intraperitoneally injected into mice for 42 days. The tumor volume and weight of nude mice treated with the high dose of NCTD was significantly smaller than that of control group (*p* < 0.05, Figure 5A–C). To further investigate the effect of NCTD on programmed cell death in the tumor tissue of xenograft mice, Terminal Deoxynucleotidyl Transferase dUTP Nick end Labeling (TUNEL) assay was performed. The results showed that the number of TUNEL-positive cells was markedly increased in the tumor tissue of NCTD-treated mice group compared to that of control group (Figure 5D). Collectively, these results suggest that NCTD effectively suppresses OSCC growth and induces programmed cell death in vivo. Then, to assess the toxicity of NCTD in vivo, the body weights of the mice were measured every week for 42 days. We found that the body and organ (liver and kidney) weights of the mice were not altered by NCTD administration (Figure 5E,F). In addition, histopathological changes of liver or kidney were not observed in NCTD-treated group compared to the control group (Figure 5G). Therefore, these results demonstrate that NCTD has a biocompatible effect without significant side effects in vivo.

## 3. Discussion

In recent decades, growing interest has been focused on natural products as chemotherapeutic or chemopreventive drugs, and they are becoming an important research area for drug discovery [15]. Recent studies have shown that cantharidin has an anticancer activity by inducing programmed cell death in various types of cancers [26,27]. However, there are some severe limits in the application of cantharidin due to its toxicity to gastrointestinal and urinary tracts [28,29]. Unlike cantharidin, NCTD has been demonstrated as a potential anticancer agent without side effects. However, the anticancer activity of NCTD toward OSCC has not been fully elucidated. In this study, we observed a decrease in survival and an increase in programmed cell death in HSC-3, HN22, and HSC-4 OSCC cell lines with various concentrations of NCTD (Figure 1, Figure 2 and Appendix A). To our best knowledge, this is the first report to show that NCTD induced programmed cell death in OSCC. Several reports have demonstrated that NCTD regulates signaling pathways playing critical roles in carcinogenesis of various cancers [21,30,31]. A study by Huang et al., demonstrated that NCTD induces programmed cell death by dose-dependently suppressing the phosphorylation of AKT and NF-κB in MDA-MB-231 [21]. The exposure to NCTD significantly elevated programmed cell death by the increases in the phosphorylation of ERK1/2, JNK, and p38 MAPK in colorectal cancer cells [32]. Sun et al., reported that NCTD treatment of HepG2 hepatocellular carcinoma cells reduced cell growth through inhibition of c-Met/mTOR signaling [33]. Using two breast cancer cell lines, estrogen receptor (ER)-HS-578T and ER+MCF-7 cells, Yang et al., suggested the potential involvement of MAPK and STAT pathways in NCTD-induced programmed cell death [34]. Our results showed that NCTD significantly induced the activation of p38 MAPK and decreased the phosphorylation of STAT3, whereas NCTD showed no apparent effect on the activation of AKT, mTOR, and ERK in HSC-3 and HN22 OSCC cells (Figure 3). p38 MAPK consists of four isoforms (p38 MAPK α, β, γ, and δ) and can be activated at different levels by diverse stimuli, which in turn trigger the networks of substrates being phosphorylated and impinge on the cellular response [35]. There is evidence that strong p38 MAPK activation is likely to engage programmed cell death, whereas lower levels of p38 MAPK activity tend to be associated with cell survival [36]. In particular, NCTD as chemical stimuli has been known to affect the phosphorylation of p38 MAPK in certain cancer cells [37]. In investigating the mechanism by which NCTD promotes programmed cell death, we first figured out that NCTD effectively activated p38 MAPK in both OSCC cell lines (Figure 3). In addition, the p38 MAPK inhibitor (SB203580) blocked NCTD-mediated programmed cell death (Figure 4). Similarly, it was previously demonstrated that the SB203580 attenuated NCTD-induced cell death of breast cancer cells [34]. On the contrary to our results, SB203580 failed to block cell death in NCTD-treated HeLa cells [38] and did not impair the effect of NCTD on the viability of human umbilical endothelial cells (HUVECs) [39]. These results suggest that the role of p38 MAPK could be different depending on the type of cancer during NCTD-induced programmed cell death. Constitutively, the activation of STAT3 is well known to play an essential role in the development of multiple cancers, including OSCC, where its hyper-activation up-regulates the transcription of cyclin D1, survivin, and Bcl-xL [8]. The activation of STAT3 has been thought to provide important diagnostic and prognostic information in OSCC [7]. The pivotal role of STAT3 in tumorigenesis has promoted a campaign in drug discovery to identify a number of small molecules compounds that directly inhibit the activity and function of STAT3 for use in cancer treatment and prevention [40]. In hepatocellular carcinoma cells, NCTD reversed IL-6-induced epithelial-mesenchymal transition process and inhibited STAT3 phosphorylation like JSI-124, a selective inhibitor of STAT3 [41]. Although NCTD exhibited dephosphorylation of STAT3 as an NCTD-induced signaling event related to programmed death of OSCC cells, STAT3-overexpressing cells could not reduce the protein level of cleaved PARP in NCTD-treated OSCC cells (Figure 3 and Appendix A). Taken together, our data indicates that the activation of p38 MAPK might be a critical signaling pathway in NCTD-induced programmed cell death in human OSCC.

Since cantharidin has been known for undesirable toxic side effects, we assessed the toxicity of NCTD in vivo [28]. Our data showed that NCTD did not change both body and organ weights and affect the histology of liver and kidney (Figure 5E–G), indicating that NCTD has negligible histological toxicity. These results agreed with the previously finding that NCTD can act as a kind of nontoxic demethylating drug, and its drug safety is supported by preclinical assessment, including acute toxicity, subchronic toxicity, hemolysis testing, intravenous stimulation, and injection anaphylaxis in mice [42,43]. These suggest that NCTD is a biocompatible agent without severe side effects.

## 4. Materials and Methods

### 4.1. Cell Culture and Chemical Treatment

Human OSCC cell lines HN22 and HSC-3 were obtained from Dankook University (Cheonan, Korea) and Hokkaido University (Hokkaido, Japan), respectively. Both cell lines were cultured in Dulbecco’s modified Eagle’s medium supplemented with 10% fetal bovine serum and antibiotics at 37 °C in a 5% CO2 incubator. All experiments were performed after the cells reached 50–60% confluence. NCTD was purchased from Sigma-Aldrich Chemical Co. (St. Louis, MO, USA) and was treated at concentrations ranging from 0 to 30 μM for 6–48 h. SB203580 (a p38 MAPK inhibitor) was purchased from Calbiochem (San Diego, CA, USA). Each chemical was dissolved in dimethyl sulfoxide (DMSO), aliquoted, and stored at −20 °C.

### 4.2. Trypan Blue Exclusion Assay

HSC-3, HN22, and HSC-4 cells were treated with different concentration- and time-dependent manners of NCTD and cell viability was determined using trypan blue staining (Gibco, Paisley, UK). Cells were stained with 0.4% trypan blue solution and viable cells were counted using a hemocytometer.

### 4.3. Live/Dead Assay

The effects of NCTD on cell death in OSCC cell lines were evaluated using a live/dead viability/cytotoxicity assay (Life Technologies, Grand Island, NY, USA). The polyanionic dye calcein-AM is retained in live cells, producing an intense green fluorescence through intracellular esterase activity. Ethidium homodimer-1 enters dead cells with damaged membranes and binds to nucleic acids, producing bright red fluorescence. Briefly, cells were stained with 2 μM calcein-AM and 4 μM ethidium homodimer-1 and incubated for 30 min at RT. Cells were analyzed under a fluorescence microscope (Leica DM5000B, Leica Microsystems, Wetzlar, Germany) with the appropriate excitation and emission filters.

### 4.4. Western Blot Analysis

Proteins were extracted from cell pellets by homogenization with a radioimmunoprecipitation assay (RIPA) buffer (EMD Millipore, Billerica, CA, USA) and the protein concentration of each sample was measured using a DC Protein Assay Kit (Bio-Rad Laboratories, Hercules, CA, USA). After normalization, equal amounts of protein were separated by sodium dodecyl sulfate-polyacrylamide gel electrophoresis and transferred to Immunoblot polyvinylidene difluoride membranes (Pall Corporation, Port Washington, NY, USA). The membranes were blocked with 5% skim milk at RT for 2 h, incubated with the specific primary antibodies, and probed with corresponding horseradish peroxidase (HRP)-conjugated secondary antibodies (GTX213110 for anti-Rabbit and GTX213111 for anti-mouse). Rabbit anti-human polyclonal antibodies against cleaved PARP (1:3000; #9541), cleaved caspase-3 (1:1000; #9664), p-STAT3 (1:1000; #9145), STAT3 (1:2000; #4904), p-p38 (1:1000; #9212), p38(1:2000; #9211), p-AKT (1:1000; #9271), p-ERK1/2(1:1000; #9101), and p-mTOR (1:1000; #2971) were purchased from Cell Signaling Technology, Inc., (Charlottesville, VA, USA). Mouse anti-human monoclonal antibodies against β-actin (1:3000; SC-47778) were obtained from Santa Cruz Biotechnology, Inc., (Santa Cruz, CA, USA). The immunoreactive proteins were detected by SuperSignal West Pico Chemiluminescent Substrate (sc-2048; Santa Cruz Biotechnology), and membranes were exposed to x-ray film. To ensure equal protein loading, each membrane was stripped and reproved with anti-β actin antibody. The densitometric analysis of western blotting was quantified using ImageJ software (version 1.51k, NIH, Bethesda, MD, USA).

### 4.5. DAPI Staining

Nuclear morphological changes of apoptotic cells were detected by DAPI fluorescence dye (Sigma-Aldrich, Louis, MO, USA). Briefly, cells were fixed with 100% methanol at RT for 10 min, deposited on slides, and stained with 2 μg/mL DAPI solution. Changes in cell morphological features during programmed cell death were observed under a fluorescence microscope.

### 4.6. Annexin V/PI Staining

Programmed cell death was evaluated by double staining with annexin- fluorescein isothiocyanate (FITC) and PI, following manufacturer’s instructions for Annexin V apoptosis assay kit (BD Biosciences, Franklin Lakes, NJ, USA). Annexin V+/PI-(early-stage apoptosis) and Annexin V+/PI+(late-stage apoptosis) regions were analyzed using a fluorescence-activated cell sorting (FACS) caliber (Becton-Dickinson) and calculated with Cell Quest software (BD Biosciences, San Jose, CA, USA).

### 4.7. Cell Cycle Analysis

Cells were fixed with 70% ethanol overnight at 4 °C and re-suspended in phosphate buffered saline (PBS) containing final concentration of 20 µg/mL RNase A and PI (P4170, Sigma-Aldrich) for 15 min at 37 °C. DNA contents were detected using FACS caliber and the relative DNA content was calculated with Cell Quest software.

### 4.8. Construction of STAT3 Over-Expression Vector and Transient Transfection

The open reading frame of human STAT3 (NM_139276) was amplified from cDNA that was synthesized in HSC3 cells using the specific primers of the gene (primer sequence; *STAT3* sense 5′-GAT ATC ATG GCC CAA TGG AAT CAG-3′, with an included EcoRV site, *STAT3* anti-sense 5′-GAT ATC TCA CAT GGG GGA GGT AGC-3′, with an included EcoRV site), and then cloned into pGEM T-easy vector (Promega, Madison, WI, USA). The STAT3 was confirmed by sequence analysis. Finally, the gene was cloned into the multi cloning site of pcDNA3.1 (+) vector (Invitrogen, San Diego, CA, USA). HSC-3 and HN22 cells were transfected by two kinds of vector constructs (pcDNA3.1; pcDNA3.1-STAT3) by Lipofectamine 3000 transfection reagent (Life Technologies, Carlsbad, CA, USA), according to the manufacturer’s instruction, respectively.

### 4.9. Nude Mouse Xenograft Assay

Seven-week-old BALB/c nu/nu male mice were purchased from NARA Biotech (Pyeongtaek, Korea). All mice were handled according to Institutional Animal Care and Use Committee (IACUC) guidelines approved by CHA University (IACUC approval number: 180154). HSC3 cells were subcutaneously injected into the flanks of the mice. Approximately 10 days after incubation (day 0), vehicle control (PBS) or NCTD (2.5 and 5 mg/kg/day) were intraperitoneally administrated to tumor bearing mice five times per week for 42 days. Tumor volume and body weight were measured once a week. The tumor volumes were measured along the two diameter axes with calipers to allow calculation of tumor volume using the following formula: V = π/6{(D + d)/2}^3^, where D and d were the larger and smaller diameters, respectively.

### 4.10. TUNEL Assay

Paraffin-embedded tumor tissues were analyzed using a TUNEL in situ apoptosis detection kit (Dead-End Colorimetric TUNEL system, Promega). Briefly, paraffin-embedded sections were deparaffinized and rehydrated. The sections were incubated with proteinase K for 15 min at RT, the endogenous peroxidase was blocked with 0.3% hydrogen peroxide for 5 min. The digoxigenine-dUTP end labeled DNA was detected using an anti-digoxigenin peroxidase antibody followed by peroxidase detection with 0.05% DAB containing 0.02% hydrogen peroxide. The sections were counterstained with methyl green, and the brown-colored apoptotic bodies in the tumor sections from control and NC-treated mice were counted using a DFC550 digital camera.

### 4.11. Histopathological Examination of Organs

Mice organs (liver and kidney) were fixed in 10% neutral buffered formalin. Tissue sections were cut at a thickness of 4 μm and stained with hematoxylin and eosin (H&E). Histopathological changes were analyzed under a DFC550 digital camera (Leica).

### 4.12. Statistical Analysis

Technical, as well as biological, triplicates of each experiment were performed. Comparison between two groups was performed by Student’s *t* test. For multiple-groups comparisons, one-way ANOVAs analysis were applied to determine the significance of differences between the control and treatment groups using by SPSS 22 (SPSS, Chicago, IL, USA); values of *p* < 0.05 were considered statistically significant.

## 5. Conclusions

In conclusion, this study provides first evidence that NCTD has an anticancer activity in both OSCC cell lines and tumor-xenograft mice. The activation of p38 MAPK could be a critical signaling kinase in NCTD-mediated programmed cell death towards OSCC. Therefore, our study provides a scope for developing NCTD as a chemotherapeutic agent against OSCCs.

## Figures and Tables

**Figure 1 ijms-20-03487-f001:**
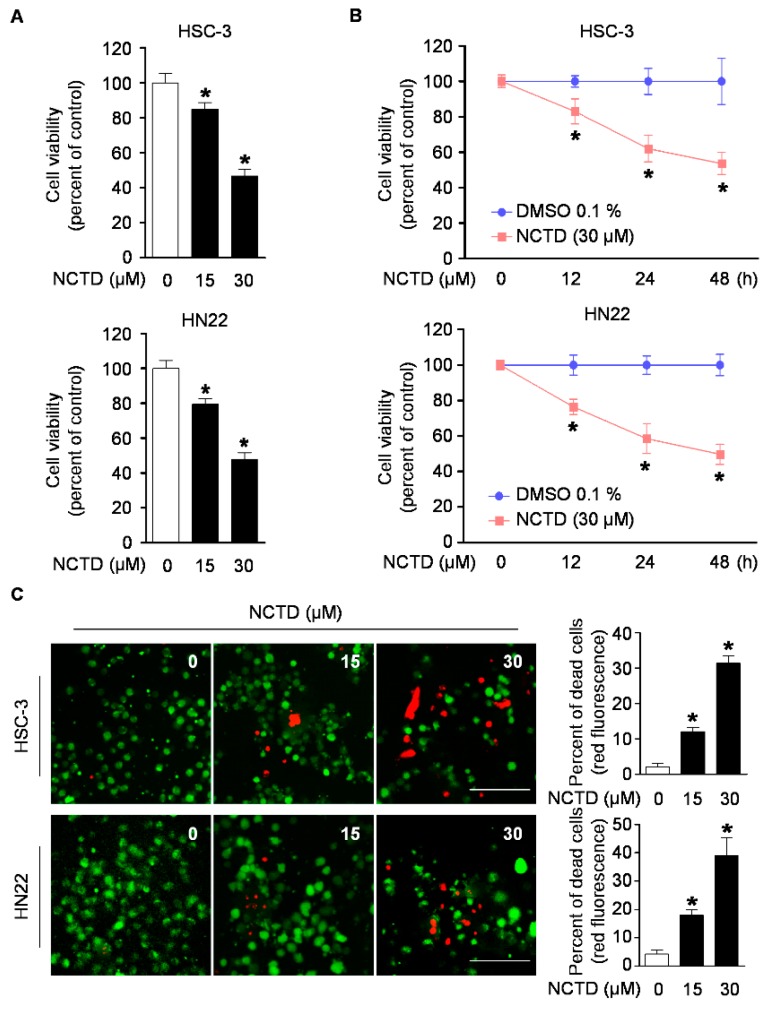
Effects of Norcantharidin (NCTD) on cell viability in human oral squamous cell carcinoma (OSCC) cell lines. HSC-3 and HN22 cells were treated with dimethyl sulfoxide (DMSO) or certain concentrations of NCTD for 48 h or 30 μM NCTD for 12, 24, or 48 h. (**A**,**B**) Cell viability was analyzed using a trypan blue exclusion assay. (**C**) Cytotoxic effect of NCTD was detected by a live/dead assay kit. For the fluorescence microscope images, live cells were stained with Calcein AM (green) and dead cells were stained with EthD-1 (red) (scale bar, 100 µm). The percentage of dead cells was quantified. Graphs represent the mean ± SD of three independent experiments, and significance compared with the control group is indicated (*).

**Figure 2 ijms-20-03487-f002:**
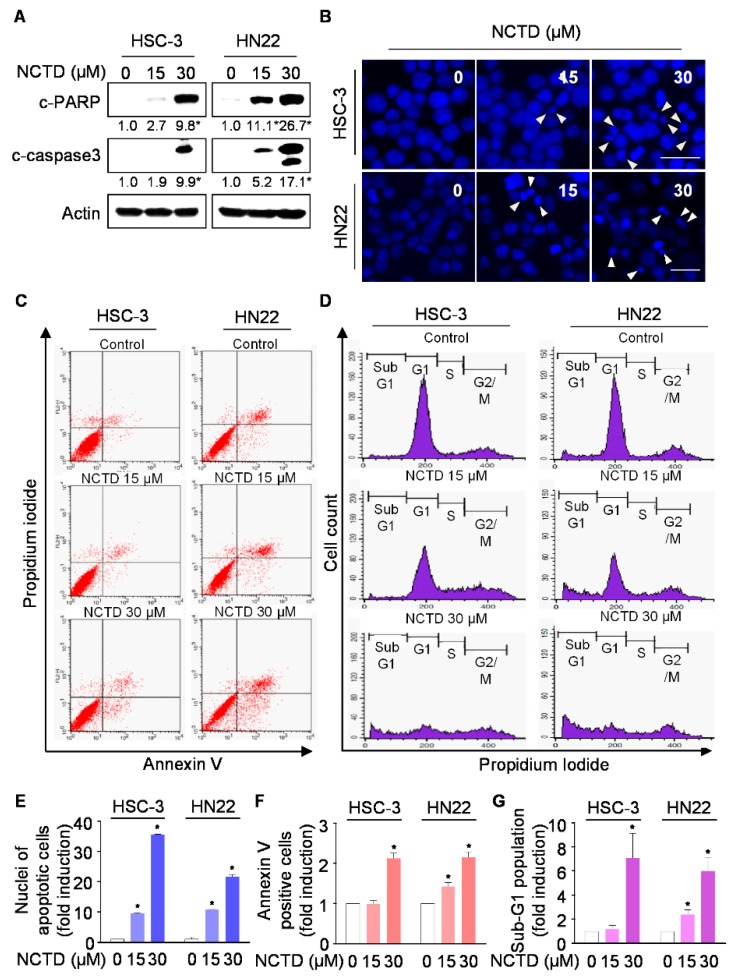
Effects of NCTD on programmed cell death. HSC-3 and HN22 cells were treated with DMSO or certain concentrations of NCTD for 48 h. (**A**) The cell lysates were analyzed by western blotting to detect the cleavages of caspase 3 and poly (ADP-ribose) polymerase (PARP). Data represent the mean of triplicate experiments. *, *p* < 0.05 is compared with the control group. (**B**) Nuclear morphology was detected by 4-6-Diamidino-2-Phenylindole (DAPI) staining, showing chromatin condensation and nuclear fragmentation (indicated by white arrows) (scale bar, 25 µm). (**C**) Apoptotic cells were detected by the annexin V/propidium iodide (PI) double-staining. (**D**) Sub-G_1_ population was analyzed by PI staining. (**E**–**G**) Quantifications of nuclei of apoptotic cells, annexin V-positive cells, and sub-G_1_ population were calculated, respectively. Graphs represent the mean ± SD of three independent experiments, and significance compared with the control group is indicated (*).

**Figure 3 ijms-20-03487-f003:**
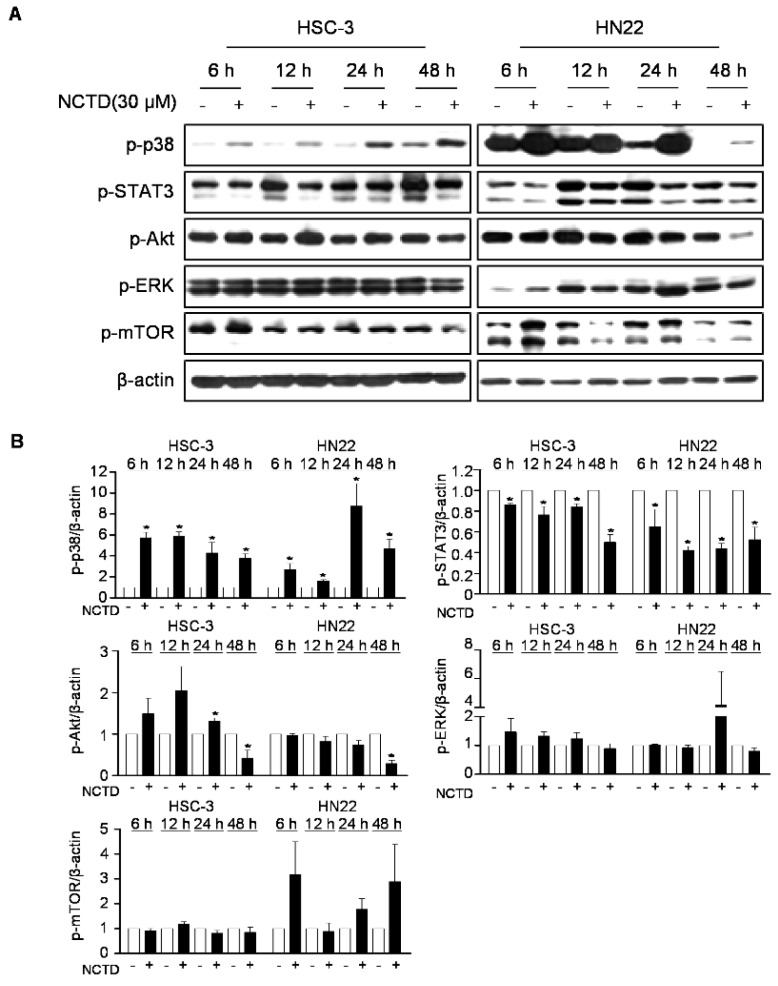
Effects of NCTD on oncogenic intracellular signaling pathways. Both cell lines were treated with 0 or 30 μM for 6, 12, 24, or 48 h. (**A**) Phosphorylated forms of p38 mitogen-activated protein kinase (MAPK), signal transducer and activator of transcription (STAT)3, AKT, extracellular signal-regulated kinase (ERK), and mammalian target of rapamycin (mTOR) were measured by western blotting. (**B**) The graph represents the mean ± SD of three independent experiments, and significance compared with the control group was indicated (* *p* < 0.05).

**Figure 4 ijms-20-03487-f004:**
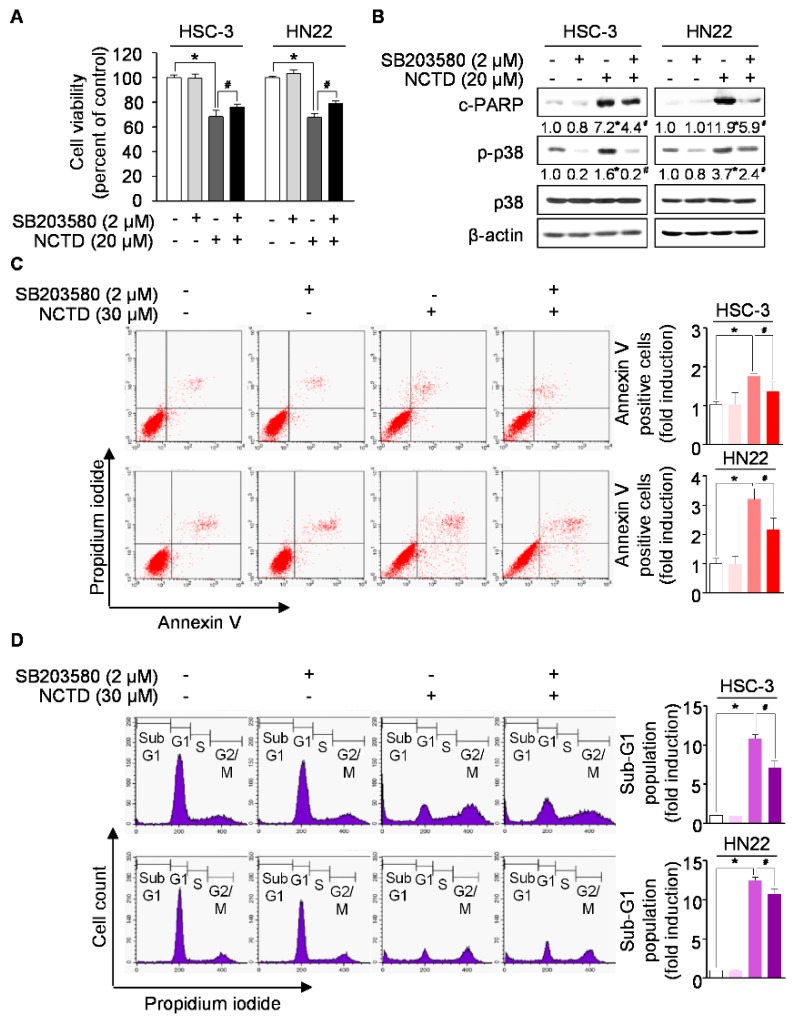
The role of p38 MAPK on NCTD-induced programmed cell death. HSC-3 and HN22 cells were pretreated with a p38 MAPK inhibitor (2 μM SB2035280) for 1 h, and certain concentrations of NCTD were added for 48 h. (**A**) Cell viability was analyzed by a trypan blue exclusion assay. (**B**) Western blotting was performed to detect the protein levels of cleaved PARP, p-p38, and p38. (**C**) Apoptotic cells were detected by the annexin V/PI double-staining. (**D**) Sub-G_1_ population was analyzed by PI staining. The graph represents the mean ± SD of three independent experiments, and significance compared with the control group was indicated (* *p* < 0.05). Significance compared with the NCTD-treated group was indicated (^#^
*p* < 0.05).

**Figure 5 ijms-20-03487-f005:**
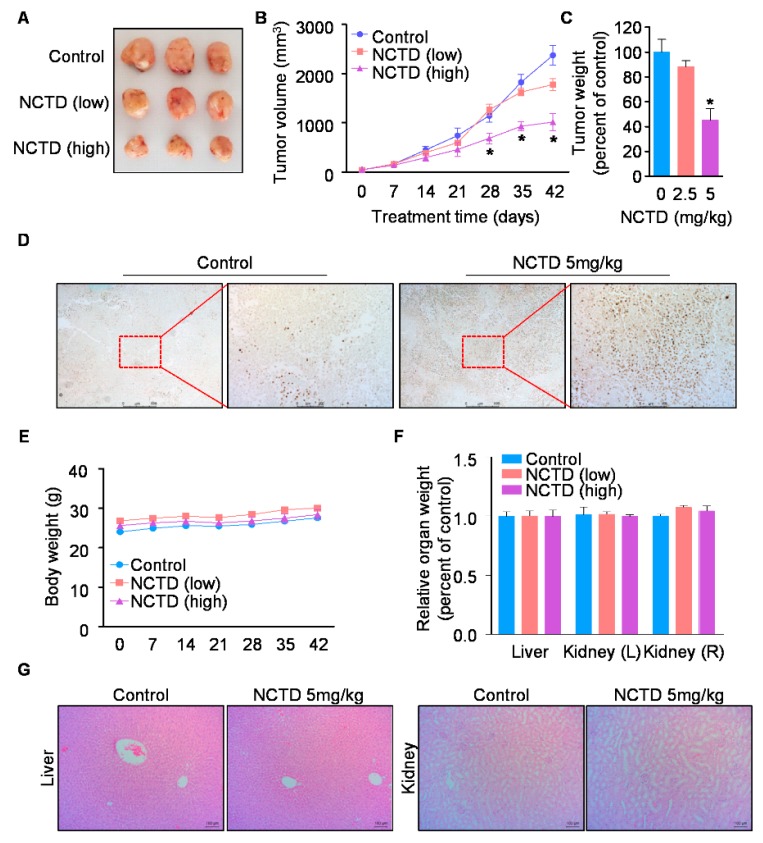
Effects of NCTD on tumor growth in a nude mouse xenograft model bearing HSC-3 cells. Athymic nude mice bearing HSC-3 cells were treated with vehicle control or NCTD (2.5 or 5 mg/kg) for 42 days (*n* = 4/group). (**A**) Representative images of tumors of mice after mice sacrificed. Tumor volume (**B**) and tumor weight (**C**) were measured. The graphs represent the mean ± S.E. of triplicate experiments. *, *p* < 0.05 is compared with control group. (**D**) Programmed cell death was detected in tumor tissues using a Terminal Deoxynucleotidyl Transferase dUTP Nick end Labeling (TUNEL) assay (original magnification, ×50). The red boxes represented cropped fields at ×200 magnification. Body weight (**E**) or organ weights (**F**) were measured. (**G**) Organ tissue samples (left panel: liver; right panel: kidney) were subjected to histopathological analysis using hematoxylin and eosin (H&E) staining (original magnification, ×100).

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
