# Peer review of "Contribution of p38 MAPK Pathway to Norcantharidin-Induced Programmed Cell Death in Human Oral Squamous Cell Carcinoma"

_ijms, 2019, doi:10.3390/ijms20143487_

Round 1

Reviewer 1 Report

In this interesting work, Ahn et al. investigated the effects of Norcantharidin, a small chemically synthesized molecule with anticancer properties, in human oral squamous cell carcinoma (OSCC) cell lines, both in vitro and in vivo. In particular, several assays were used (blue exclusion assay, live/dead assay, western blotting, DAPI staining, flow cytometric analysis, TUNEL assay, and immunohistochemistry) to investigate apoptotic effect and molecular targets of Norcantharidin. The results were interesting, suggesting that this molecule could be a potential anticancer drug candidate against OSCC. Furthermore, the use of an inhibitor of p38 MAPK partially attenuated the apoptotic effects, suggesting an important role of this molecule in Norcantharidin-induced apoptosis.

The study is on a timely subject in view of increasing interest about the identification prognostic molecular markers, and to identify new therapeutic targets in cancers.

The techniques utilized were appropriate and described with plenty details. This is a well-designed study with rigorous methods. The discussion is well-balanced and the statements are supported by the data.

I suggest adding some considerations in the Introduction section related to the prognosis of OSCC. In particular, the unchanging survival in patients with OSCC underscores the need for better prognostic tools, such as the adding of Depth of invasion and Extranodal extension in the 8th Edition AJCC Cancer Staging Manual, as recently reported [1].

[1] Mascitti M, et al. American Joint Committee on Cancer staging system 7th edition versus 8th edition: any improvement for patients with squamous cell carcinoma of the tongue? Oral Surg Oral Med Oral Pathol Oral Radiol. 2018 Nov;126(5):415-23.

Author Response

Reviewer 1

In this interesting work, Ahn et al. investigated the effects of Norcantharidin, a small chemically synthesized molecule with anticancer properties, in human oral squamous cell carcinoma (OSCC) cell lines, both in vitro and in vivo. In particular, several assays were used (blue exclusion assay, live/dead assay, western blotting, DAPI staining, flow cytometric analysis, TUNEL assay, and immunohistochemistry) to investigate apoptotic effect and molecular targets of Norcantharidin. The results were interesting, suggesting that this molecule could be a potential anticancer drug candidate against OSCC. Furthermore, the use of an inhibitor of p38 MAPK partially attenuated the apoptotic effects, suggesting an important role of this molecule in Norcantharidin-induced apoptosis.

The study is on a timely subject in view of increasing interest about the identification prognostic molecular markers, and to identify new therapeutic targets in cancers.

The techniques utilized were appropriate and described with plenty details. This is a well-designed study with rigorous methods. The discussion is well-balanced and the statements are supported by the data.

I suggest adding some considerations in the Introduction section related to the prognosis of OSCC. In particular, the unchanging survival in patients with OSCC underscores the need for better prognostic tools, such as the adding of “Depth of invasion” and “Extranodal extension” in the 8th Edition AJCC Cancer Staging Manual, as recently reported [1].

 [1] Mascitti M, et al. American Joint Committee on Cancer staging system 7th edition versus 8th edition: any improvement for patients with squamous cell carcinoma of the tongue? Oral Surg Oral Med Oral Pathol Oral Radiol. 2018 Nov;126(5):415-23.

à Thank you for your thoughtful comments. It will make our introduction more obvious. As you suggested, we added this reference to the Introduction section.

Reviewer 2 Report

In the present study, the authors studied the potential anticancer effects of Norcantharidin (NCTD) against human oral squamous cell carcinoma (OSCC) in vitro and in vivo. To prove their hypothesis, they used different assays including trypan blue exclusion assay, live/dead assay, western blotting, DAPI staining, flow cytometric analysis, TUNEL assay, and immunohistochemistry. Authors found that NCTD inhibited OSCC cell growth and increased the number of dead cells through the cleavages of casepase-3, increase in apoptotic morphological changes and increase in annexin V-positive cells or sub-G1 population of cells. Via p38 MAPK inhibitor, they also reported that NCTD significantly activated p38 MAPK pathway, but inactivated STAT3 pathway. A (SB203580) partially attenuated NCTD-induced programmed cell death (apoptosis) in OSCC cell lines. NCTD strongly suppressed tumor growth in the tumor xenograft. Based on these findings, authors suggested NCTD as potential anticancer drug candidate for the treatment of OSCC. OSCC is a relatively fatal malignancy, which has recently been increased in incidence, and it is therefore interesting to search for potential novel therapeutic targets. This is an interesting study, and the topic itself is interesting and falls within the journal’s scope, however, there are a few comments regarding this study that authors should address before publication:

1)      NCTD is an analogue of cantharidin and both are isolated from mylabris (or blister beetles). Cantharidin effects on OSCC (including apoptosis) have already been investigated (check for e.g. Su et al., 2016). Therefore, authors should provide comprehensive argument regarding the novelty of the present work, and why their study should be considered a novel adding to the previous reports.

2)      The text requires English-language revision. I recommend authors to seek proofreading from English-speaking colleague/service.

3)      Authors used various concentrations of NCTD (0, 15, or 30 μM) for 12, 24, 48 hrs. What is the rationale of selecting such concentrations and time slots?

4)      It would be interesting to check the NCTD effect on other caspases as well (e.g. caspase-7 and 9). Furthermore, cantharidin was shown to reduce pro-caspase-12 protein expression. I recommend to check for that too.

5)      I also recommend to authors to provide a schematic representation of the proposed scenario of NCTD on OSCC. This would make it easier to follow.

Author Response

Reviewer 2

In the present study, the authors studied the potential anticancer effects of Norcantharidin (NCTD) against human oral squamous cell carcinoma (OSCC) in vitro and in vivo. To prove their hypothesis, they used different assays including trypan blue exclusion assay, live/dead assay, western blotting, DAPI staining, flow cytometric analysis, TUNEL assay, and immunohistochemistry. Authors found that NCTD inhibited OSCC cell growth and increased the number of dead cells through the cleavages of casepase-3, increase in apoptotic morphological changes and increase in annexin V-positive cells or sub-G1 population of cells. Via p38 MAPK inhibitor, they also reported that NCTD significantly activated p38 MAPK pathway, but inactivated STAT3 pathway. A (SB203580) partially attenuated NCTD-induced programmed cell death (apoptosis) in OSCC cell lines. NCTD strongly suppressed tumor growth in the tumor xenograft. Based on these findings, authors suggested NCTD as potential anticancer drug candidate for the treatment of OSCC. OSCC is a relatively fatal malignancy, which has recently been increased in incidence, and it is therefore interesting to search for potential novel therapeutic targets. This is an interesting study, and the topic itself is interesting and falls within the journal’s scope, however, there are a few comments regarding this study that authors should address before publication:

1)      NCTD is an analogue of cantharidin and both are isolated from mylabris (or blister beetles). Cantharidin effects on OSCC (including apoptosis) have already been investigated (check for e.g. Su et al., 2016). Therefore, authors should provide comprehensive argument regarding the novelty of the present work, and why their study should be considered a novel adding to the previous reports.

 à Thank you for your kind comments. As you mentioned, cantharidin effects on OSCC have already been investigated suggesting that cantharidin can be a good drug candidate for OSCC. However, several studies already showed that the use of cantharidin was limited by its toxicity to gastrointestinal and urinary tracts. Norcantharidin as a synthetic cantharidin derivative, is a demethylated form of cantharidin and is relatively free from side effects, including bone marrow suppression and gastrointestinal and urinary tract toxicity. Thus, we thought that it can be a valuable study to investigate the anticancer effect of norcantharidin in OSCC.

2)      The text requires English-language revision. I recommend authors to seek proofreading from English-speaking colleague/service.

 à As you suggested, we checked the English language using a professional editing service to ensure that it is correct, clear and concise. (E-World Editing: www.eworldediting.com)

3)      Authors used various concentrations of NCTD (0, 15, or 30 μM) for 12, 24, 48 hrs. What is the rationale of selecting such concentrations and time slots?

à Several studies showed that they used 10-30 μM for 24 and 48 hrs to inhibit cell growth and induce apoptosis in several cancer cell lines. (Ref. 4-5). Based on the previous literatures, we selected 15 and 30 μM for its concentration and our cell viability assay showed that its IC50 values were 30.37 μM and 29.83 μM in HSC-3 and HN22 cells, respectively. We also used 6, 12, 24 and 48 hrs time slots to understand the underlying mechanism of NCTD-induced apoptosis in both cell lines.

4)      It would be interesting to check the NCTD effect on other caspases as well (e.g. caspase-7 and 9). Furthermore, cantharidin was shown to reduce pro-caspase-12 protein expression. I recommend to check for that too.

à Thank you for you good comments. We also basically agree with your comments to check caspase-7 and caspase 9. On the other hand, caspase 7 is one of executioner caspases like caspase 3, so we don’t think we need to check the expression level of caspase 7. In the present study, we did not want to investigate whether NCTD-induced apoptosis is in an intrinsic or extrinsic pathway, but we focused on the involvement of protein kinases in NCTD-induced apoptosis. Thus, we did not check the expression level of caspase 9. However, to identify whether NCTD affects an intrinsic apoptotic pathway or an extrinsic pathway is very interesting, so we will explore it for the further direction.

à The study finding that cantharidin was shown to reduce pro-caspase 12 protein expression is quite related to endoplasmic reticulum (ER) stress. Thus, we don’t think we need to check the expression level of pro-caspase 12 at this time. However, your comment is quite interesting, so we will explore the effect of NCTD on ER stress including pro-caspase 9 during NCTD-induced apoptosis in OSCC cell lines in the future.

5)      I also recommend to authors to provide a schematic representation of the proposed scenario of NCTD on OSCC. This would make it easier to follow.

à Thank you for your kind comments. As you suggested, we made a schematic representation of the proposed scenario of NCTD on OSCC.
